# Adequate Boron Supply Modulates Carbohydrate Synthesis and Allocation in Sugarcane

**DOI:** 10.3390/plants14050657

**Published:** 2025-02-21

**Authors:** Jorge Martinelli Martello, Murilo de Campos, Carlos Antônio Costa do Nascimento, Ariani Garcia, Miriam Büchler Tarumoto, Gabriela Ferraz de Siqueira, Patrick H. Brown, Carlos Alexandre Costa Crusciol

**Affiliations:** 1Lageado Experimental Farm, Department of Crop Science, College of Agricultural Sciences, São Paulo State University (UNESP), P.O. Box 237, Botucatu 18610-307, SP, Brazil; jorgemartinelli@gmail.com (J.M.M.); murilodecampos83@gmail.com (M.d.C.); ariani_garcia@hotmail.com (A.G.); miriam.tarumoto@unesp.br (M.B.T.); gaferrazsiq@gmail.com (G.F.d.S.); 2Department of Crop Science, Luiz de Queiroz College of Agriculture (USP-ESALQ), University of São Paulo, Piracicaba 13418-900, SP, Brazil; cacnascimento@usp.br; 3Department of Plant Sciences, College of Agricultural and Environmental Sciences, Davis 1 Shields Ave., Davis, CA 95616, USA; phbrown@ucdavis.edu

**Keywords:** sugarcane nutrition, sugarcane varieties, *Saccharum* spp., nutrient solution, sugarcane root and shoot development

## Abstract

Boron (B) is an essential and widely studied element in plants. Due to B dynamics in highly weathered soils, its concentration is generally low. Among other benefits, B interacts with calcium pectate, promotes stability on cellular membrane, and influences directly on plant nutrients uptake and non-structural metabolites synthesis. In sugarcane (*Saccharum* spp.) crop, adequate B supply has been associated with juice quality and yield of stalks and sugar and its response on adequate B concentration on commercial fields can differ greatly even into a group of varieties recommended for the same production environment. In this context, the authors aimed to assess the effects of B availability on sugarcane root and shoot development, nutrient status, and carbohydrate synthesis and allocation in two sugarcane varieties recommended for the same production environment using hydroponic solution. The experimental design was completely randomized and consisted of four treatments and four replicates. The treatments comprised two sugarcane varieties (RB867515 and RB92579) and two B concentrations (0.05 and 0.5 mg L^−1^) considered deficient and adequate, respectively, for plant development. Carbohydrate partitioning, nutrient concentrations in various plant parts, and growth and morphological parameters were evaluated. Under adequate B supply, the total concentrations of reducing sugars and sucrose increased 67 and 20% in RB867515 and 30 and 20% in RB92579, respectively, whereas starch decreased by 27% for both varieties. Adequate B supply increased the concentrations of all elements in all plant organs, except for N and K in leaves, and improved most yield and morphological parameters. Principal component analysis correlated the higher carbohydrates concentration and yield parameters with the variety RB92579, whereas the highest concentration of most nutrients was mainly associated with the variety RB867515, especially under adequate B supply. The main influence of adequate B supply was on carbohydrate synthesis. Although the sugarcane varieties responded differently to B availability, their biometric parameters were enhanced by adequate B supply. These results emphasize the need for B fertilization, regardless of the sugarcane variety’s susceptibility to B deficiency.

## 1. Introduction

Boron (B) is involved in multiple structural and physiological processes in plants. The importance of B for plant cell wall structure, cell membranes, and vegetative and reproductive growth was comprehensively summarized by Brown et al. [1]. Due to the low mobility of B in plants, symptoms of deficiency initially occur in young organs; new leaves appear twisted, with folds and creases. In more advanced stages of B deficiency, symptoms evolve to death of the apical meristem and, eventually, the entire plant [2]. In acidic soils, soil organic matter (SOM) is the main source of B for plants [3,4]. Regardless of the crop, B deficiency specifically inhibits the growth of tissues and reproductive structures [1].

Sugarcane (*Saccharum* spp.) is one of the most important crops grown in Brazil due to the country’s large-scale sugar and ethanol production and strong participation in domestic and overseas markets [5]. However, the intensification and expansion of sugarcane cultivation to areas with soils of low fertility and low SOM content have reduced the average yield of Brazilian production [6,7]. The low B bioavailability due to its dynamics in these highly weathered and/or sandy soils, which its absorption may be reduced in periods of drought or be leached in periods of high rainfall, and the lower absorption induced by high K saturation mainly in areas under vinasse application, further contributes to reducing the average yield of sugarcane [8]. These facts have been described in the literature [9,10] and are frequently observed by agronomic consultants and researchers. For instance, Mellis et al. [11] reported that applying 2 kg ha^−1^ of B at the bottom of the planting row increased the average stalk yield among 11 sites by approximately 12 tons, and Crusciol et al. [8] concluded that increases in stalk and sugar yields in soils of different textures were highly correlated with B application.

The importance of adequate B fertilization has motivated more detailed studies of the specific effects of B in plants, such as the influence of B on root elongation and architecture [12], plant respiration and carbohydrate metabolism [13], biological nitrogen fixation [14], induction of aluminum toxicity and salinity tolerance [15,16,17,18,19], antioxidant capacity [20], and acquisition and physiological interaction of other nutrients, such as potassium (K) and phosphorus (P) uptake [21] and the relationship with calcium pectate through crosslinking of the microfibrillar network, improving the tensile properties of cell walls [22]. Most of these symptoms may be an indirect effect of B deficiency on cellular and non-structural carbohydrate metabolism due to the role of B on membrane stability and functionality [1].

Essentially, reducing sugars (RS) are highly reactive monosaccharides (glucose and fructose) ready to be oxidized as a supplier of carbon skeletons and energy for metabolic and physiological processes. Conversely, sucrose is a non-reducing sugar composed of the carbonyl group present in fructose and glucose forming a stable glycosidic bond used as a transport sugar. Finally, starch is a temporary and non-transportable carbohydrate storage synthesized when the rate of sucrose export is lower than the photosynthetic rate [23,24]. In sugarcane, B nutrition is particularly important because its deficiency leads to starch accumulation in leaves, which may reduce CO_2_ assimilation by the inhibition of photosynthesis, increase phenolic compounds accumulation, and decrease nitrate uptake rate caused by the reduced transcription level of root plasma membrane H^+^-ATPase [13,25]. Also, the starch accumulation in leaves impairs phloem loading of photoassimilates, reduces sucrose accumulation in stalks, and impacts sugarcane quality and profitability [26,27]. The mechanism and intensity of these effects, however, may vary between varieties of a species [1,2,28], and these differences have not been adequately documented, which raises questions on the susceptibility to B deficiency for different sugarcane varieties, even when they are recommended for the same production environment. For instance, the variety RB867515 exhibits visual symptoms and reduced yield under B deficiency differently from the variety RB92579 which does not present such visual symptoms and severe yield reduction [29].

According to Brown et al. [1], B absorption occurs by passive diffusion through intermediate channels located at the lipid bilayer on which facilitate B absorption and can be saturated even at low B concentrations. On the other hand, due to the role of B in non-structural carbohydrate synthesis and translocation [29,30], Pfeffer et al. [31] demonstrated in sunflower (*Helianthus annuus*) roots that the B absorption mechanism is dependent on metabolic processes that can be modulated depending on the B concentration. Accordantly, we hypothesized that the feedback mechanism to balance carbohydrate synthesis and consumption and its influence on plant morphological parameters, such as root length, thickness, and surface area, may vary for different sugarcane varieties and B availability scenarios and explain the distinct response of plants under deficiency or adequate B supply.

In the field, B is essentially applied as solid or liquid fertilizers at planting or topdressing [8,11] and eventually supplemented as foliar solution [32]. On the other hand, due to the difficulty in controlling B deficiency and/or sufficiency, in accessing sugarcane roots adequately in field trials, and considering that the authors’ intention is not provide information on B management, but study the plants metabolic response, the present study was set up hydroponically and aimed to understand how two varieties of sugarcane recommended for the same production environment could respond differently regarding B availability, considering the concentration of nutrients and the synthesis and allocation of carbohydrates in roots and shoots and whether such effects influence the yield of plant biomass.

## 2. Results

### 2.1. Carbohydrate Concentrations and Allocation

Regardless of the plant organ, the concentrations of carbohydrates differed significantly among the treatments. The adequate B supply increased the total concentrations of reducing sugars (RS) and sucrose by 67 and 20%, respectively, in RB867515 and 30 and 20%, respectively, in RB92579 (Figure 1a–c). Conversely, the total concentration of starch was 27% higher in both varieties under B deficiency.

The concentration of RS in the stalks were reduced by 5 and 10% in RB867515 and RB92579, respectively, under adequate B supply, while increased by 5% and 12% in leaves. The concentration of RS in the roots increased under adequate B supply in both sugarcane varieties but maintained similar percentage distribution to plants under B deficiency (Figure 1a). The sucrose concentrations in the leaves, stalks, and roots were higher under adequate B supply than under B deficiency in both varieties (Figure 1b). Unlike RS and sucrose, the starch concentration increased under deficient B supply, especially in roots (Figure 1c). Despite the lowest total concentration of RS and sucrose and the highest concentration of starch in plants under B deficiency, their percentage distribution was similar, regardless of the variety.

### 2.2. Nutritional Status

Adequate B supply increased the leaf concentrations of P, Ca, and B regardless of sugarcane variety (Figure 2). In the roots, the concentrations of N, P, K, Ca, and B were significantly higher under adequate B supply than under deficient B supply (Figure 2), except for P in RB92579 (Figure 2b).

In the stalks, adequate B supply increased the concentrations of all elements except P in RB92579 (Figure 2b). The concentrations of all elements except B were higher in RB867515 than in RB92579 (Figure 2).

The leaf concentrations of P and Ca were highest in RB867515, while the leaf concentration of B was highest in RB92579 (Figure 2b,d,e).

### 2.3. Plant Growth, Root Morphological Parameters, and Biomass Production

Biomass production (shoot dry matter mass) and morphological parameters differed significantly among the treatments, with the greatest improvements under adequate B supply (Figure 3a–f). There was no influence of variety on the leaf dry matter (LDM), and stalk dry matter (SDM) was higher in RB92579 than in RB867515. The B application increased the shoot dry matter (LDM + SDM) by 27 and 15% in RB867515 and RB92579, respectively, root dry matter (RDM) by 19 and 23% (Figure 3a); and plant high (PH) by 12% and 4% (Figure 3b). For both varieties, root length (RL) was approximately 23% higher under adequate B supply than under deficient B supply (Figure 3b). Conversely, the root diameter (RDi) was higher under deficient B supply and no effect was observed on SDi (Figure 3c). The internode length (IL) performed quite similarly to PH, and the root surface area (RSA) was 17% (RB867515) and 19% (RB92579) higher when B was adequately supplied (Figure 3d). Although plant tillering was greater in RB92579 than in RB867515, no influence of B levels on the number of leaves per plant was observed (Figure 3e,f).

### 2.4. Varietal Differences

PCA provided a better overview of the parameters and the relationships between varieties and B supply (Figure 4). Carbohydrate concentrations (pink vectors), nutrient concentrations (green vectors), and yield and morphological parameters (orange vectors) differed between varieties (Figure 4). Higher carbohydrate concentrations were associated with RB92579, whereas the concentrations of all elements, except the leaf concentrations of B and N, were correlated with RB867515. Biomass yield and morphological parameters were strongly associated with RB92579, except for RL and RSA. These associations with variety were particularly evident under adequate B supply. Conversely, under deficient B supply, RDi was correlated with RB867515. Thus, the PCA revealed differences between the varieties, even though both exhibited better performance under adequate B supply.

## 3. Discussion

### 3.1. Carbohydrate Concentrations and Allocation

B is essential for carbohydrate synthesis and transport, which explains the increase in the concentration of RS and sucrose when an adequate rate of B was supplied [13]. Studies involving B application in Citrus have revealed the role of B on carbohydrates transport evidencing higher accumulation in leaves and lower accumulation on roots under deficient than adequate B scenarios differently to our findings [13,33,34]. Our study was carried out for 100 days, which corresponds to a period of rapid cellular expansion. During this period, sucrose is transported and enzymatically hydrolyzed by acidic invertases, and the resulting hexoses are rapidly used for plant growth. As a result, a positive feedback mechanism between carbohydrate synthesis and consumption is favored [29,30]. In this scenario, an adequate B supply stimulates carbohydrate synthesis in higher-yielding plants to balance the consumption of carbohydrates by the sink organs (stalk and roots).

The highest concentration of sucrose was observed in stalks under adequate B supply. B application can enhance the transport of non-structural carbohydrates, especially sucrose. The loading of sucrose into phloem is catalyzed by a proton gradient and an H^+^/sucrose co-transporter established by the H^+^-ATPase located on the plasma membrane of cells. However, B deficiency can lead to a decline in transcription levels of the plasma membrane H^+^-ATPase [25,35]. Any change in H^+^-ATPase activity may impair sucrose transport in B-deficient plants. Additionally, B can bind adjacent hydroxyl groups to form stable complexes with organic molecules in the cell walls of vascular plants [36] These properties explain the importance of B in stabilizing the pectin fraction in the cell wall, the formation of conducting tissues, and the adequate flow of sugars from source to sink organs.

Starch is the major storage of carbohydrates in plants and tends to concentrate in the roots, particularly in stressed plants [37]. Under B deficiency, tissue growth is inhibited, and the demand for carbohydrate synthesis is reduced [38] which may explain the enhanced concentration of starch at all plant organs.

### 3.2. Nutritional Status

The efficiency of nutrient absorption varies directly with root length and thickness, and therefore the lower nutrient concentrations under B deficiency were due to the reduced length and surface area of roots. Nutrients may also interact in higher plants such that one nutrient negatively or positively affects the absorption, redistribution, or function of another nutrient [39]. For example, B-N interactions in stalks and roots are attributed to the ability of B to increase the activity of the enzyme N-reductase [25]. B-P and B-K interactions reflect the effects of B on membrane permeability and the absorption of ions by plant cells through stimulation of the enzyme ATPase and induction of beta K channels in roots [2,35]. The possible mechanisms by which this control is exerted include direct interaction of B with polyhydroxy components of the membrane and the elevation of endogenous levels of auxins [40]. B-Ca interactions are attributed to the role of B in Ca translocation and subsequent solubilization in tissues [41]. Under B deficiency, plants tend to accumulate Ca in insoluble form as a cell wall component due to the highly similar functions of B and Ca in plant metabolism [41].

PCA showed that B absorption was more efficient in RB92579 than in RB867515 under adequate B conditions, resulting in higher stalk and leaf concentrations of B. This difference between varieties may be related to the ability to restrict B accumulation in shoots and roots due to the activity of boric acid transporters, which accumulate B independently of the applied concentration. Varieties that are highly tolerant of B deficiency are more efficient in the upregulation of boric acid/borate efflux transporters, which confer tolerance to B deficiency and toxicity [42]. Although leaf concentrations of B were lower under B deficiency, they remained within the adequate range for sugarcane development [43,44].

### 3.3. Plant Growth and Morphological Parameters

The inhibition or reduced apical cell elongation of roots and shoots observed in the present study is one of the earliest responses to B deficiency in higher plants [42,45] In both varieties, most yield parameters were reduced under its deficiency. Lu et al. [34] also observed the morphological parameters were reduced by B deficiency in Citrus. These effects are likely associated with possible impairment of the expression of specific enzymes (i.e., xyloglucan endotransglycosylases/hydrolases (XTHs), expansins, pectin methylesterases, polygalacturonases, and pectate lyases) that are responsible for increasing cell length [13,41]. B availability can also affect the synthesis of important hormones. Eggert and von Wirén [40] observed that the synthesis of abscisic acid (ABA) and indole-3-acetic acid (IAA) was reduced under an adequate supply of B, whereas cytokinin and gibberellin synthesis were strongly increased, which influenced root and shoot elongation. Also, B deficiency may induce chlorophyll degradation, compromising photosynthetic efficiency. When the photosynthetic apparatus is compromised, the electron transport chain and CO_2_ assimilation are reduced, which will impact biomass yield [42].

In both varieties under B deficiency, RDi increased, whereas PH and IL decreased. According to PCA, the increase in RDi was associated with variety RB867515 which presented higher RDi than in RB92579 even under adequate B supply. Because B is essential for meristematic tissue development and cell wall structure, B deficiency can cause poor cell wall formation in the roots and inhibit cell expansion, ultimately resulting in newly divided cells of abnormal size and shape [28]. The thickening of roots under B deficiency is unfavorable for plant growth since fine roots are primarily responsible for the absorption of water and nutrients [2,46]. This fact may be one of the points linked to the genetic characteristics of the varieties that justify a better performance of RB92579 in relation to RB867515 in several aspects as discussed, including the greater susceptibility to B deficiency of RB867515 in relation to RB92579.

Although the effects of B supply on the sugarcane varieties differed, the results clearly demonstrated the essential role of B on sugarcane carbohydrates metabolism, nutrients uptake, and morphological parameters. Efforts are currently underway to test several B sources and application strategies to make B management more accessible for sugarcane producers.

## 4. Materials and Methods

### 4.1. Experimental Design and Varieties

The experiment was conducted in a greenhouse with a climate-controlled environment. An internal heating/cooling and air circulation system maintained the temperature between 21 and 30 °C. The experimental design was completely randomized and consisted of four treatments and four replicates. The treatments comprised two sugarcane varieties (RB867515 and RB92579) and two B concentrations (0.05 and 0.5 mg L^−1^) in the nutrient solution. These B concentrations are considered deficient and adequate, respectively, for plant development [47]. The varieties were selected by considering information from sugarcane consultants and variety reports [29]. Variety RB867515 is very susceptible to soil B deficiency and exhibits visual symptoms and reduced yield. By contrast, variety RB92579 shows no visible symptoms of B deficiency, and its yield is unaffected by an inadequate B supply [29].

### 4.2. Seedling Management and Nutrient Solution

Cane cuttings for seedling preparation were harvested from a commercial nursery. Cane cuttings of approximately 3 cm containing one vegetative bud were extracted from the upper third of the cane to ensure greater sprouting uniformity. The cuttings were germinated in plastic trays containing coarse sand as substrate and irrigated daily with deionized water. The trays were distributed on benches at room temperature, which ranged between 25 and 28 °C. Thirty days after emergence, healthy seedlings were selected, fixed in Styrofoam plates, and transferred to plastic pots (four sugarcane seedlings per pot) containing 14 L of a nutrient solution under constant aeration.

The modified version of the nutrient solution proposed by Furlani and Furlani [45] was used The nutrient concentrations in mg L^−1^ were as follows: 138 N-NO_3_^−^ (Ca(NO_3_)_2_·4H_2_O; NH_4_NO_3_; KNO_3_); 20 N-NH_4_^+^ (NH_4_NO_3_; (NH_4_)_6_Mo_7_O_24_); 16 P (KH_2_PO_4_); 141 K (KCl; K_2_SO_4_; KNO_3_; KH_2_PO_4_); 151 Ca (Ca(NO_3_)_2_·4H_2_O); 17 Mg (MgSO_4_·7H_2_O); 56 S (K_2_SO_4_; MgSO_4_·7H_2_O; ZnSO_4_·7H_2_O; CuSO_4_·5H_2_O); 0.04 Cu (CuSO_4_·5H_2_O); 3.6 Fe (Rexolim M48 (6.5% Fe chelated with EDDHMA)); 0.5 Mn (MnCl_2_·4H_2_O); 0.08 Mo ((NH_4_)_6_Mo_7_O_24_); 0.15 Zn (ZnSO_4_·7H_2_O); 33 Cl (KCl; MnCl_2_·4H_2_O); and 0.05 or 0.5 B (H_3_BO_3_ (Sigma-Aldrich*^®^*, St. Louis, MO, USA); deficient and adequate, respectively).

To avoid any possible saline effects on the initial growth of the seedling roots, the nutrient solution was diluted to one-third ionic strength in the first week and half ionic strength in the second week. In the third week, the nutrient solution was used at full ionic strength. The maximum variation of the solution volume was 5%, and the volume was adjusted with deionized water. The pH of the nutrient solution in each pot was monitored daily and maintained at 5.5 ± 0.5 by adjustment with 0.1 M HCl or NaOH [48]. The nutrient solution was replaced weekly, and the plants were maintained in contact with the nutrient solution for 100 days.

### 4.3. Growth Parameters

After 100 days of plant growth in the nutrient solution, non-destructive analyses were performed. A graduated ruler was used to measure plant height (PH) from the root-shoot junction to the auricular region of leaf ^+^1 or TVD (top visible dewlap leaf) [49]. The stalk diameter (SD) in the middle third of the stalk was determined with the aid of a digital caliper. The average internode length (IL) was calculated by dividing PH by the number of leaves (NL) [50]. The number of tillers (NT) was counted.

### 4.4. Root Morphological Parameters

After the growth parameter analyses, the plants were separated into stalks, leaves, and roots. Three subsamples of the roots were extracted from the root-shoot junction to the end of the root system, stored in 30% alcohol solution in a universal collector with a capacity of 100 mL, and refrigerated until further analysis. Root length, surface area, and diameter (RL, RSA, and RDi, respectively) were determined using a scanner coupled to a computer running WinRhizo software version 3.8-b (Regent Instruments Inc., Québec, QC, Canada) as described by Tennant [51].

### 4.5. Plant Dry Matter Production

The stalks, leaves, and roots of the plants in each experimental unit were dried to constant weight in an oven with forced-air circulation at 65 °C and weighed to determine the yields of root dry matter (RDM), stalk dry matter (SDM), and leaf dry matter (LDM). The samples of the plant root system used in the morphological determinations were also oven-dried at 65 °C to determine the dry matter yield, which was added to the RDM.

### 4.6. Nutrient Concentrations

After dry matter determination, the plant material was ground in a Wiley mill, and the concentrations of nitrogen (N), phosphorus (P), potassium (K) calcium (Ca), and B in the vegetative tissues were determined according to Malavolta et al. [52]. P, K, and Ca were extracted by nitroperchloric acid digestion and determined by atomic absorption spectrophotometry. N was extracted using H_2_SO_4_ (sulfuric acid) solution and the Kjeldahl distillation method and determined by titration. B was determined by colorimetry. Before the experiments, the concentration of B in the reserve tissue of the seedlings was determined.

### 4.7. Carbohydrate Concentrations

The partitioning of carbohydrates between plant organs was represented by total sugars (glucose + fructose + sucrose) and reducing sugars (RS, glucose + fructose) and determined by fractionation technique using a high-performance liquid chromatography (HPLC) on a Shimadzu model 10A chromatograph (Kyoto, Japan) with an RID-10A refractive index detector and model LC-10AD isocratic pump with mobile phase flow of 0.6 mL m^−1^ and purified water. A leaf sample of 1.0 g of and 8.0 mL of purified water were weighed and incubated in a metabolic bath model MA095 (Marconi) at 60 °C for 40 min under constant agitation and centrifugation at 12,000 rpm to separate the solid part. The supernatant was filtered through Millipore polyvinyl decafluoride (PVDF) membrane with a porosity of 0.22 m and 13.0 mm in diameter. The samples were then injected into the liquid chromatograph to obtain the chromatograms and compared with predefined concentration sucrose, glucose, and fructose (reducing sugars) pattern curves. Sucrose and reducing sugars were calculated by comparing their areas with their standards and multiplied by the dilution of each sample.

The starch was extracted by homogenizing 200 mg of DM in 42 mL of deionized water. The pH of the suspension was subsequently adjusted to 4.8 using sodium acetate buffer. Then, 10 µL of the enzyme α-amylase was added to the sample, followed by stirring in a water bath at 90 °C for 2.5 h. After cooling, 100 µL of amyloglucosidase was added, and the samples were stirred in a water bath for an additional 2.5 h. Acidity was neutralized by adding 4 M NaOH, and the volume was adjusted to 250 mL. A 5-mL aliquot was then transferred to a 100-mL flask and adjusted to a final volume of 100 mL with water. The starch concentration in the dry matter mass was determined according to Somogyi-Nelson [50,51], and readings were acquired in a spectrophotometer at 535 nm.

#### Statistical Analysis

Data were submitted to analysis of variance (ANOVA) and the least significant difference (LSD) test at 5% significance to compare means. Principal component analysis (PCA) was performed to test the relationships between variables, and only loadings with an absolute value greater than 60% of the maximum coefficient in each principal component (PC) were considered [52]. In PCA, each variable is represented by a vector, and the length of each vector indicates the strength of its contribution. The relative importance of each variable can be estimated from the perpendicular projection of each sample to its respective vector. Statistical analyses were carried out using JMP statistical software version 10.0.0 (SAS Institute Inc., Cary, NC, USA).

## 5. Conclusions

Although the plants in this study were adequately nourished, B can also interact positively with other essential nutrients. The B supply in RB867515 impacted positively on root length and increased the uptake of most nutrients while RB92579 was more efficient in absorbing and accumulating B and N.

During this sugarcane phenological stage (100 days), the main influence of adequate B supply is on non-structural carbohydrate synthesis. Despite the significant increase in the concentration of carbohydrates in different parts of the plants (root, stalk and leaf), their relative distribution among plants in the same variety under or without B deficiency is quite similar.

The two sugarcane varieties responded differently to B availability, however, adequate B supply enhanced the biometric parameters of both varieties, supporting the need for B fertilization regardless of its susceptibility to B deficiency.

## Figures and Tables

**Figure 1 plants-14-00657-f001:**
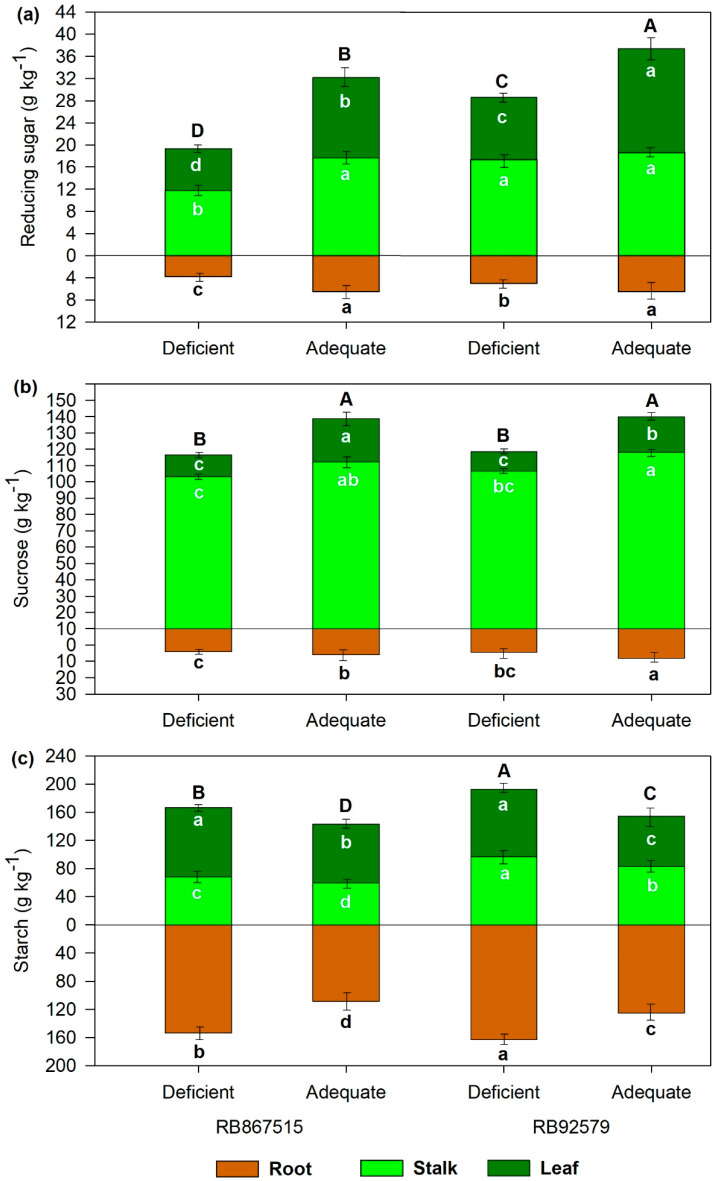
Carbohydrate partitioning (**a**) reducing sugars, (**b**) sucrose, and (**c**) starch in separated plant’s part as a function of B levels and sugarcane varieties. Different letters into the same plant organs indicate significant differences by Student’s *t*-test at *p* ≤ 0.05. Capital letters represent total concentration and bars represent the standard deviation (n (composite samples of roots, shoots, and leaves) = 4).

**Figure 2 plants-14-00657-f002:**
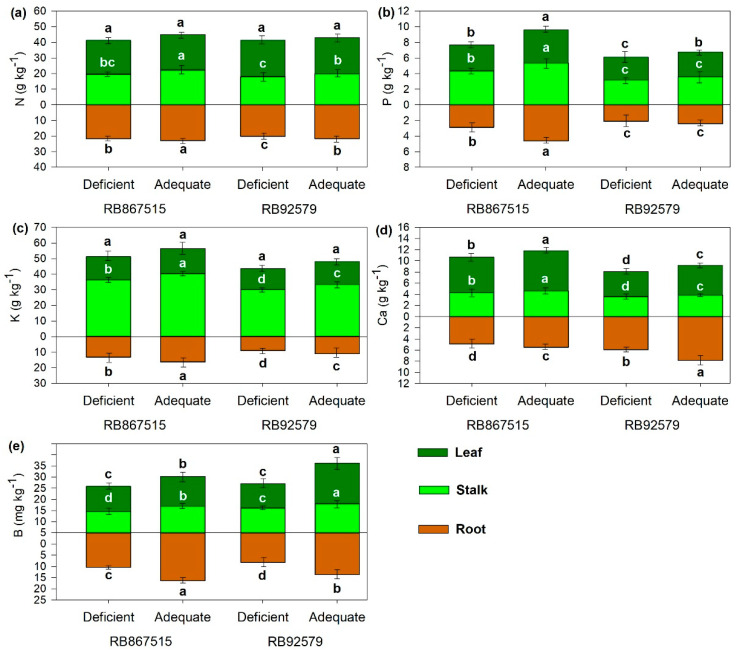
Roots, stalk, and Leaf concentration of (**a**) N, (**b**) P, (**c**) K, (**d**) Ca, and B (**e**) as a function of B levels and sugarcane varieties. Different letters indicate significant differences by Student’s *t*-test at *p* ≤ 0.05 and bars represent the standard deviation (n (composite samples of roots, shoots, and leaves) = 4).

**Figure 3 plants-14-00657-f003:**
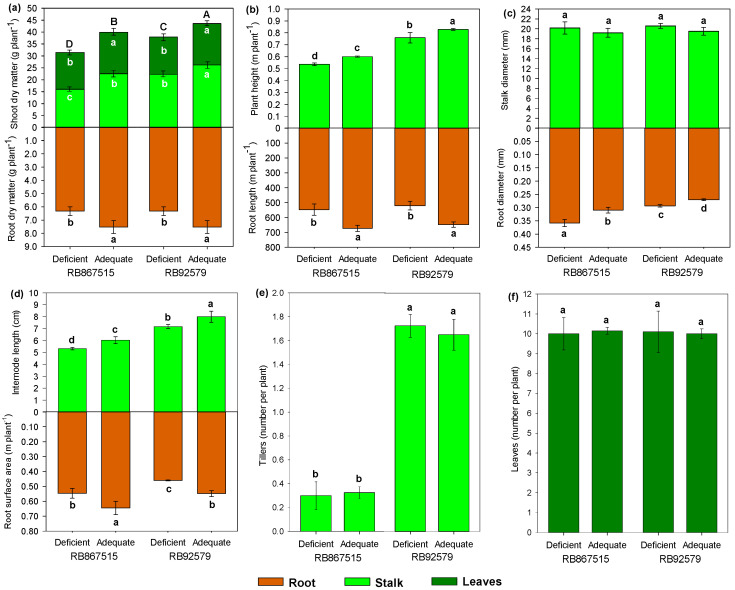
Biometrics of (**a**) root and shoot dry matter, (**b**) root length and plant height, (**c**) root and stalk diameter, (**d**) root surface area and internode length, (**e**) number of tillers, and (**f**) number of leaves per plant as a function of B levels and sugarcane varieties. Different letters indicate significant differences by Student’s *t*-test at *p* ≤ 0.05. Capital letters represent total concentration and bars represent the standard deviation (n (composite samples of roots, shoots, and leaves) = 4).

**Figure 4 plants-14-00657-f004:**
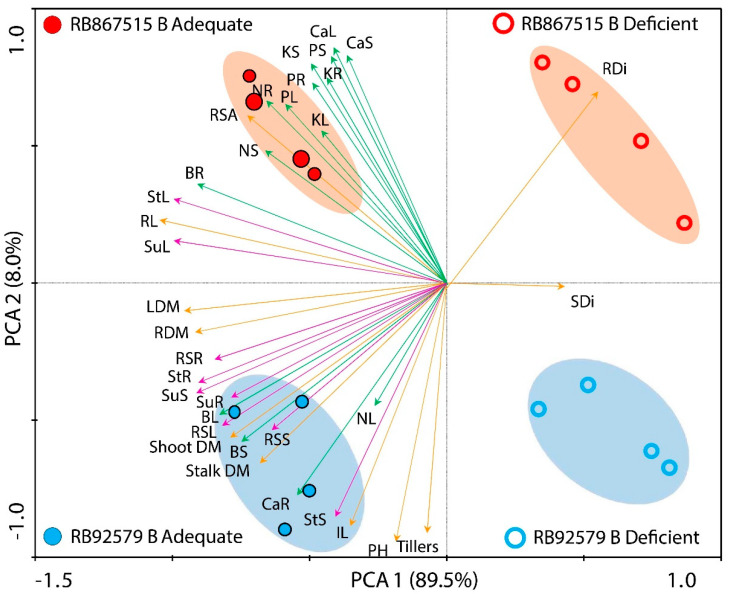
Principal Component Analysis (PCA) between varieties (RB867515 (red circles) and RB92579 (blue circles)) and B levels (Adequate (fulfilled circles) and Deficient (empty circles)). The vectors presented by the colors green, orange, and pink represent the nutritional, growth, and carbohydrates parameters, respectively. Root diameter (RDi), root length (RL), root surface area (RSA), root dry matter (RDM), leaf dry matter (LDM), stalk diameter (SDi), stalk dry matter (Stalk DM), number of tillers (Tillers), plant height (PH), internode length (IL), nitrogen concentration in the root (NR), nitrogen concentration in the leaf (NL), nitrogen concentration in the stalk (NS), phosphorus concentration in the root (PR), phosphorus concentration in the leaf (PL), phosphorus concentration in the stalk (PS), potassium concentration in the root (KR), potassium concentration in the leaf (KL), potassium concentration in the stalk (KS), calcium concentration in the root (CaR), calcium concentration in the leaf (CaL), calcium concentration in the stalk (CaS), boron concentration in the root (BR), boron concentration in the leaf (BL), boron concentration in the stalk (BS), reducing sugar concentration in the root (RSR), sucrose concentration in the root (SuR), starch concentration in the root (StR), reducing sugar concentration in the leaf (RSL), sucrose concentration in the leaf (SuL), starch concentration in the leaf (StL), reducing sugar concentration in the stalk (RSS), sucrose concentration in the stalk (SucS), starch concentration in the stalk (StS). 4 plants per experimental unit were used.

## Data Availability

Data available in a publicly accessible repository.

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
