# Peer review of "Adequate Boron Supply Modulates Carbohydrate Synthesis and Allocation in Sugarcane"

_plants, 2025, doi:10.3390/plants14050657_

Round 1
Reviewer 1 Report (Previous Reviewer 3)
Comments and Suggestions for Authors
The authors have answered all my concerns.
Author Response
Thank you for your comments and suggestions. They certainly improved the quality of the paper.
Reviewer 2 Report (New Reviewer)
Comments and Suggestions for Authors
Dear authors,
The manuscript presents an interesting and significant contribution to the field, offering valuable insights into the study's subject matter. However, several issues need to be addressed to enhance its overall quality and impact. These include improving the clarity and coherence of the discussion, ensuring accurate citation and referencing of tables and figures, providing a more detailed explanation of key findings, and justifying statistical analyses more effectively. Addressing these concerns will strengthen the manuscript and make it more accessible to a broader audience.
1. Abstract Improvement: The abstract should begin with a clear and concise problem statement. Please ensure that the research problem is explicitly stated at the beginning.
2. Hypothesis Clarity: While the hypothesis is mentioned in the introduction, it should be more concise and clearly stated to provide a direct and well-defined research premise.
3. Novelty & Importance: At the end of the introduction, please provide a brief explanation of the study’s novelty and significance to highlight its contribution to the field.
4. Statistical Analysis: Since the study involves two factors (sugarcane varieties and boron supply), a two-way ANOVA would be a more appropriate statistical approach to analyze the differences and interactions between these factors.
5. Citation Format (Lines 55–58): In the literature review, when referring to an author's findings, it is better to mention the author's name before the citation number. For example, instead of “[11] reported,” write “Mellis et al. [11] reported.”
6. Table Citation Issue (Line 88): Table 1 is cited in the text, but no such table appears in the manuscript. Please carefully review the entire manuscript to ensure all citations correspond accurately to existing figures and tables.
7. Reducing Sugars Measurement: Please clarify the significance of measuring reducing sugars in this study and how it relates to the research objectives.
8. Table Citation Issue (Line 109): Table 1 is referenced again, but it is missing from the manuscript. Please verify and correct this inconsistency.
9. Figure Citation Issue: Figure 2d is not cited in the text. Ensure all figures are referenced appropriately within the manuscript.
10. Figure Caption (Line 122): In the figure caption, specify what a replicate represents. Does it refer to an individual plant, a sample type, or another specific unit?
11. Missing Data on Plant Yield (Line 125): Plant yield is mentioned, but no corresponding data is presented in the manuscript. Please include relevant data or clarify its omission.
12. Accurate Figure Citations (Line 126): Ensure that figure parts are accurately cited where the relevant data is explained in the text.
13. Comparative Analysis (Lines 131–132): The sentence states that root length (RL) was approximately 23% higher under adequate boron supply than under deficient supply (Figure 3b). However, it is also essential to compare the two sugarcane varieties and indicate which variety had a greater root length.
14. Figure Captions & Error Bars: In figure captions, clearly specify what the bars represent (e.g., mean ± SD or mean ± SE). Additionally, Figures 1 and 2 lack error bars, while Figure 3 includes them. Please justify this inconsistency.
15. PCA Justification: The discussion requires more justification for the PCA results. Several correlations need explanation. For example, why was root diameter (RDi) higher under boron deficiency in RB867515 compared to boron adequacy? Similarly, why was stalk diameter (SDi) higher under boron deficiency in RB92579? Further clarification of these and other relevant parameters is needed.
16. Discussion Revision: The discussion should be rewritten to improve clarity and coherence. I suggest structuring it to compare the study’s findings with previous research, highlighting similarities and differences with appropriate explanations.
17. Materials & Methods Detail: The Materials and Methods section should be more descriptive to ensure the study is reproducible. Provide sufficient detail on experimental conditions, measurements, and methodologies used.
Comments on the Quality of English Language
The overall quality of English in the manuscript is average and requires moderate revision to enhance clarity and coherence. Some sentences are too long and complex, making it difficult to grasp the intended meaning. I suggest simplifying these sentences while maintaining their original intent to improve readability. Additionally, improving the logical flow between ideas will help ensure that the manuscript is more engaging and easier for readers to follow. A thorough language revision will enhance the overall presentation and effectiveness of the manuscript.
Author Response
Please see the attachment.

Reviewer 3 Report (New Reviewer)
Comments and Suggestions for Authors
The work is excellent and provides essential information for optimizing the management of boron (B) use in sugarcane. The results are solid, and the discussion presents an interesting dialogue with the findings. However, some aspects could be improved.
The introduction appropriately presents the importance of B for sugarcane and cites previous studies. However, it could further explore the physiological mechanisms of B and its interactions with other nutrients. Specifically, discussing the impact of B on carbohydrate metabolism, cell wall structuring, and its interaction with Ca would enrich the text and provide a better foundation for understanding the results. Additionally, in the second line of the Introduction, the citation format should be adjusted— the author's name should be mentioned before the reference number. For example, the sentence "The importance of B for plant cell wall structure, cell membranes, and vegetative and reproductive growth was comprehensively summarized by [1]" should include the author's name before the reference.
Materials and Methods: The authors state that 0.05 mg L⁻¹ of B is considered deficient and 0.5 mg L⁻¹ is adequate. It is important to specify the literature source that supports this classification. Furthermore, at several points in the methodology section, references are mentioned using only numbers. I suggest including the name of the first author before the reference number to improve readability. Nutritional Analysis: The study focuses on the role of B, but it does not include an analysis of Mg and S, and other micronutrients. Given that B interacts with several nutrients in plant metabolism, including these analyses could provide insights into possible nutritional interactions that influence the observed responses.
While the results are well described, the discussion could be improved by strengthening correlations with previous studies. Although some prior research is cited, the discussion should delve deeper into comparing the results obtained with existing literature, highlighting similarities and differences. Molecular Aspects of Boron: The role of B in sugar translocation and H⁺-ATPase activity is mentioned but not sufficiently detailed at the molecular level. Including a more in-depth discussion on the biochemical pathways involved would enhance the manuscript. Practical Implications: The study emphasizes the importance of B for sugarcane productivity, but it does not sufficiently discuss practical strategies for optimizing B management under field conditions. Providing recommendations or discussing different fertilization strategies could improve the applicability of the findings.
The manuscript presents relevant and well-structured research, with scientifically sound results and an important contribution to sugarcane nutrition management. The identified points for improvement are minor and do not compromise the validity of the study. Therefore, I recommend acceptance with minor revisions (minor review).
Round 2
Reviewer 2 Report (New Reviewer)
Comments and Suggestions for Authors
Dear authors,
Thank you for improving the manuscript. I found some improvements in the revised version but some the issues that I mentioned in my previous comments still not not resolved. I am attaching them below:
1. The abstract should begin with a clear and concise problem statement. Please ensure that the research problem is explicitly stated at the beginning.
2. While the hypothesis is mentioned in the introduction, it should be more concise and clearly stated to provide a direct and well-defined research premise. I couldn’t find improvement to the hypothesis.
3. Novelty of the study wasn’t included and my comment was ignored.
4. About statistical analysis the response is not satisfying.
Round 3
Reviewer 2 Report (New Reviewer)
Comments and Suggestions for Authors
Dear authors,
After reading-evaluating the revised manuscript and the second round comments, I found that manuscript is significantly improved. However, I suggest you to improve the introduction.
Author Response
Please see the attachment

This manuscript is a resubmission of an earlier submission. The following is a list of the peer review reports and author responses from that submission.
Round 1
Reviewer 1 Report
Comments and Suggestions for Authors
Overall, the manuscript looks good; however, significant changes are needed particularly in the result section.
Line 25-27: “Under adequate B supply-------------whereas starch decreased 27% for both varieties”, in comparison to?
Line 29-31: “Principal component ---------especially under adequate B supply.”, rephrase for better clarity.
Line 50: “soil organic matter (SOM)”; there is no need to put the full name again and again.
In the introduction, please follow similar trends for citations (e.g., see lines 68 and 70).
Line 87: “The total concentrations of RS and sucrose increased, respectively, 67 and 20%” rephrase for better clarity.
In the result section, many sentences are too confusing, please rewrite these and have a look at the overall section.
Figure 1: You can reduce the width of the bars for a better look.
Figure 2: Please re-arrange the scale values.
Line 124: unit error for the number of leaves.
Line 130: “SDM was higher in RB92579”, as compared to??? Please carefully check this issue throughout the section. The same problem is in line 132.
Line 260: B deficiency à its deficiency
In the reference section, some recent studies from the 'Plants' Journal can be added.
Comments on the Quality of English LanguageSignificant improvements are needed particularly in the result section.
Author Response
We totally appreciate your comments and suggestions that certainly improved the quality of the paper. Please, find the responses below. In the comments we have made great modifications, they were rephrased directly in the text.
Thank you very much.
Comment 1. Line 25-27: “Under adequate B supply-------------whereas starch decreased 27% for both varieties”, in comparison to?
Response 1. The change was made in the text. We rewrite “Under adequate B supply, the total concentrations of reducing sugars and sucrose were 67 and 20% higher in RB867515 and 30 and 20% higher in RB92579, respectively, whereas starch decreased 27% for both varieties” to “Under adequate B supply, the total concentrations of reducing sugars and sucrose increased 67 and 20% in RB867515 and 30 and 20% in RB92579, respectively, whereas starch decreased by 27% for both varieties”.
Comment 2. Line 29-31: “Principal component ---------especially under adequate B supply.”, rephrase for better clarity.
Response 2. The change was made in the text. We rephrased “Principal component analysis showed that RB92579 was associated with carbohydrates and yield parameters, whereas RB867515 was associated with the concentrations of most nutrients, especially under adequate B supply” to “Principal component analysis correlated the higher carbohydrates concentration and yield parameters with the variety RB92579, whereas the highest concentration of most nutrients was mainly associated with the variety RB867515, especially under adequate B supply”.
Comment 3. Line 50: “soil organic matter (SOM)”; there is no need to put the full name again and again.
Response 3. The full name was maintained only in the first citation and used only the initials for the next.
Comment 4. In the introduction, please follow similar trends for citations (e.g., see lines 68 and 70).
Response 4. Citations were standardized according to the journal´s template.
Comment 4. Line 87: “The total concentrations of RS and sucrose increased, respectively, 67 and 20%” rephrase for better clarity.
Response 4. The change was made in the text. We rephrased “The total concentrations of RS and sucrose increased, respectively, 67 and 20%, in RB867515 and 30 and 20% in RB92579 under adequate B supply” to “The adequate B supply increased the total concentrations of RS and sucrose by 67 and 20%, respectively, in RB867515 and 30 and 20%, respectively, in RB92579”.
Comment 5. In the result section, many sentences are too confusing, please rewrite these and have a look at the overall section.
Response 5. Any modifications were made in the result section to clarify it (please, see the file with modifications attached).
Comment 6. Figure 1: You can reduce the width of the bars for a better look.
Response 6. We reduced the width bars in Figure 1 from 60 to 45% for a better look.
Comment 7. Figure 2: Please re-arrange the scale values.
Response 7. The scale values were re-arranged in Figure 2.
Comment 8. Line 124: unit error for the number of leaves.
Response 8. We rearranged the panel from Figure 2 to Figure 3 for a better look and changed the legend.
Comment 9. Line 130: “SDM was higher in RB92579”, as compared to??? Please carefully check this issue throughout the section. The same problem is in line 132.
Response 9. We have made modifications in the text (please, see the file with modifications attached).
Comment 10. Line 260: B deficiency à its deficiency
Response 10. The sentence has rewritten
Comment 11. In the reference section, some recent studies from the 'Plants' Journal can be added.
Response 11. The citations below were added:
- Effects of Foliar Boron Application on Physiological and Antioxidants Responses in Highbush Blueberry (Vaccinium corymbosum L.) CultivarsReyes-Díaz M, Cárcamo-Fincheira P, Inostroza-Blancheteau CPlants 2024, Vol. 13, Page 1553 (2024) 13(11) 1553
- Effects of Exogenous Boron on Salt Stress Responses of Three Mangrove SpeciesYang J, Wei H, Liu XPlants 2025, Vol. 14, Page 79 (2024) 14(1) 79
- Optimizing Sugarcane Growth, Yield, and Quality in Different Ecological Zones and Irrigation Sources Amidst Environmental StressorsManzoor M, Khan M, Haider FPlants 2023, Vol. 12, Page 3526 (2023) 12(20) 3526
- Supplemental Silicon and Boron Alleviates Aluminum-Induced Oxidative Damage in Soybean RootsWang S, Cheng H, Wei YPlants 2024, Vol. 13, Page 821 (2024) 13(6) 821
- Understanding Ameliorating Effects of Boron on Adaptation to Salt Stress in ArabidopsisQu M, Huang X, Shabala SPlants 2024, Vol. 13, Page 1960 (2024) 13(14) 1960

Reviewer 2 Report
Comments and Suggestions for Authors
I found the paper well-written and interesting to a reader. However, several suggestions should be considered before publication:
1. The citation of literature data must be rearranged. In the present variant, the sources (at least, absolute majority) in the text are cited with authors' names (not with numbers) but, in References, they are given by numbers, in the order, as they were first mentioned in the text. This does not fit the journal's requirements and makes the checking of references more difficult.
2. I suggest that the information about two tested sugarcane varieties, namely, that they contrasted with each other by their susceptibility to boron deficiency (lines 278-282), should be included not only in Materials and Methods, but also in Introduction. This information seems essential for the proper understanding of this paper.
3. I suggest that the Conclusions should be rearranged or rewritten. In the present variant, the sentence in line 375 seems to contradict the one which follows in lines 376-377. It remains doubtful whether boron actually stimulated not only sucrose synthesis but also its allocation. This doubt is related to the fact (which the authors themselves stress in lines 376-377) that "stalk sucrose concentrations were similar to those under adequate B supply".
Author Response
We totally appreciate your comments and suggestions that certainly improved the quality of the paper. Please, find the responses below. In the comments we have made great modifications, they were rephrased directly in the text.
Thank you very much.
I found the paper well-written and interesting to a reader. However, several suggestions should be considered before publication:
Comment 1. The citation of literature data must be rearranged. In the present variant, the sources (at least, absolute majority) in the text are cited with authors' names (not with numbers) but, in References, they are given by numbers, in the order, as they were first mentioned in the text. This does not fit the journal's requirements and makes the checking of references more difficult.
Response 1. Citations were standardized according to the journal template.
Comment 2. I suggest that the information about two tested sugarcane varieties, namely, that they contrasted with each other by their susceptibility to boron deficiency (lines 278-282), should be included not only in Materials and Methods, but also in Introduction. This information seems essential for the proper understanding of this paper.
Response 2. We added this information from line 73 to 76.
Comment 3. I suggest that the Conclusions should be rearranged or rewritten. In the present variant, the sentence in line 375 seems to contradict the one which follows in lines 376-377. It remains doubtful whether boron actually stimulated not only sucrose synthesis but also its allocation. This doubt is related to the fact (which the authors themselves stress in lines 376-377) that "stalk sucrose concentrations were similar to those under adequate B supply".
Response 3. The conclusion section was rewritten (please, see the file with modifications attached).

Reviewer 3 Report
Comments and Suggestions for Authors
The authors of this manuscript investigated the effects of B on sugar synthesis and distribution, as well as element absorption in two sugarcane varieties. However, there are several issues with the manuscript that need to be addressed:
- None of the Figures include error analysis, and the Figure legends do not specify information about biological replicates of the samples.
- In Figures 2 and 3, the font size of the x-axis labels is too small, making them difficult to read.
- What is the purpose of Table 1? The significance analysis is already presented in the Figures, so why is it listed separately in a Table?
- Abbreviations should be spelled out in full when they first appear in the text.
- Panels (f) and (j) in Figure 2 would be better placed in Figure 3.
Author Response
We totally appreciate your comments and suggestions that certainly improved the quality of the paper. Please, find the responses below. In the comments we have made great modifications, they were rephrased directly in the text.
Thank you very much.
The authors of this manuscript investigated the effects of B on sugar synthesis and distribution, as well as element absorption in two sugarcane varieties. However, there are several issues with the manuscript that need to be addressed:
Comment 1. None of the Figures include error analysis, and the Figure legends do not specify information about biological replicates of the samples.
Response 1. The information about error analysis and biological replications was added in the figures.
Comment 2. In Figures 2 and 3, the font size of the x-axis labels is too small, making them difficult to read.
Response 2. The modifications on the font size were done.
Comment 3. What is the purpose of Table 1? The significance analysis is already presented in the Figures, so why is it listed separately in a Table?
Response 3. Table 1 was removed from the text.
Comment 4. Abbreviations should be spelled out in full when they first appear in the text.
Response 4. The full names were included in the first appearance and the abbreviations were used next times.
Comment 5. Panels (f) and (j) in Figure 2 would be better placed in Figure 3.
Response 5. The panels (f) and (j) in Figure 2 were replaced to Figure 3